# Enhancement of the Bioactive Compound Content and Antibacterial Activities in Curcuma Longa Using Zinc Oxide Nanoparticles

**DOI:** 10.3390/molecules28134935

**Published:** 2023-06-23

**Authors:** Munirah F. Aldayel

**Affiliations:** Department of Biological Sciences, College of Science, King Faisal University, AlAhsa 31982, Saudi Arabia; maldayel@kfu.edu.sa; Tel.: +966-597-279-936

**Keywords:** *Curcumin longa*, zinc oxide, antibacterial, nanoparticles

## Abstract

Incorporating nanoparticles into plant cultivation has been shown to improve growth parameters and alter the bioactive component compositions of many plant species, including *Curcumin longa*. The aim of the current study was to investigate the effects of foliar application of zinc oxide nanoparticles on the content of bioactive compounds and their antibacterial activities against potential bacterial pathogens. To this end, *C. longa* leaves were treated with different doses of ZnO NPs to see how this affected their bioactive component composition. The effect of different doses of ZnO NPs on the accumulation of bisdemethoxycurcumin, demethoxycurcumin, and curcumin in ethanolic extracts of *C. longa* rhizomes was evaluated using high-performance liquid chromatography (HPLC). When compared to the control treatment, foliar spraying with (5 and 40 mgL^−1^) of ZnO NPs increased bisdemethoxycurcumin, demethoxycurcumin, and curcumin levels approximately (2.69 and 2.84)-, (2.61 and 3.22)-, and (2.90 and 3.45)-fold, respectively. We then checked whether the ethanolic extracts produced from the plantlets changed in terms of their phytochemical makeup and antibacterial properties. Furthermore, the results revealed that *C. long*-ZnO NPs displayed antibacterial activity against the tested *S. aureus* and *P. aeruginosa* bacterium strains, but had a few effect against *E. coli.* The MIC for *P. aeruginosa* was 100 g/mL. The time–kill studies also revealed that ZnO NPs at 4 MIC killed *P. aeruginosa*, *Actinobacteria baumannii*, and *Bacillus* sp. after 2 h, while *S. aureus* did not grow when treated with 4 × MIC of the extract for 6 h. The strongest antibacterial activity was seen in the extract from plantlets grown without nanoparticles for *P. aeruginosa*, whereas it was seen in the extract from plantlets grown in the presence of 5 mg/L ZnO NPs for *E. coli*, *S. aureus*, and *P. aeruginosa*. These findings show that ZnO NPs are powerful enhancers of bioactive compound production in *C. longa,* a trait that can be used to combat antibiotic resistance in pathogenic bacterial species.

## 1. Introduction

Recently, increased attention has been devoted to the potential use of nanotechnology in plant biotechnology. Substances between 0.1 and 100 nm in size are referred to as nanomaterials. Plants’ responses to nanomaterials and nanoparticles vary widely depending on their size, shape, application technique, chemical properties, and physical qualities [1]. Zinc oxide nanoparticles (ZnO NPs) are a particular type of NP that has sparked the curiosity of plant researchers. Despite their phytotoxicity, the application of ZnO NPs significantly promotes plant growth [2]. Several phytotoxic effects of ZnO NPs were outlined by [2]. Interestingly, in the presence of modest concentrations of ZnO NPs, beneficial effects on plant growth were documented in several of the published studies [3,4,5]. Numerous studies have shown that ZnO NPs improve a variety of plant processes and properties, including germination rates, growth rates, yields, organ development, somatic embryogenesis, somaclonal variation, genetic transformation, and the production of secondary metabolites [6,7], while also enhancing field applications [3,8,9].

The application of metal nanomaterials can result in multiple beneficial impacts. For example, the use of zinc oxide nanoparticles increased wheat seedlings’ resistance to the effects of drought stress [10] and stimulated nickel removal by Sorghum bicolor’s metal ecotoxic potential and plant response [11]. Additionally, green-synthesized silver–zinc oxide nanocomposites from *Curcuma longa* extracts showed antioxidant, antibacterial, and antibiofilm potential against multidrug-resistant enteroaggregative *Escherichia coli* [12,13].

The incidence of microbial infections has multiplied by many orders of magnitude over the course of the last decade as a direct result of the development of multidrug resistance. In addition, one of the most critical challenges in green nanotechnology is the development of a method that is both easy on the environment and efficient in producing various metal oxide nanoparticles.

The effectiveness of zinc-oxide-nanoparticle-coated aligners as an antimicrobial agent against *Streptococcus mutans* and *Candida albicans* has recently been studied [14,15]. In that study, it was reported that the maximal antibacterial impact was not seen until 2 days after the application of the ZnO-nanocoated aligners, and that it lasted for 7 days after that. The impact on *Candida albicans* was not very significant. It seems that using ZnO-coated aligners as a method to promote antibacterial activity against *S. mutans* is a promising strategy [14,16]. Infections caused by *Acinetobacter baumannii* may occur in the blood, the urinary system, and the lungs (pneumonia), in addition to wounds located in other areas of the body. Additionally, *Acinetobacter baumannii* is able to “colonies or live within a patient without causing infections or symptoms, particularly in respiratory secretions (sputum) or open wounds.

*Curcuma longa*, a popular medicinal herb, belongs to the Zingiberaceae family [17]. Known popularly as “turmeric,” *C. longa* is a spice and coloring component that is well known for its medicinal properties [18]. Turmeric has three major curcuminoids: curcumin (diferuloyl methane), demethoxycurcumin, and bisdemethoxycurcumin [19]. Curcumin, the most important component, is responsible for turmeric’s biological benefits. Curcumin has a melting point of 184 ° C. It is water-insoluble, but soluble in ethanol and acetone [18]. Curcumin, a powerful antioxidant with anti-inflammatory, antioxidant, and anti-platelet properties, is regarded as turmeric’s most bioactive and soothing component. *S. albus* and *S. aureus* were suppressed by *C. longa* oil at doses of 1 to 5000 [20]. Turmeric contains anti-inflammatory and antibacterial properties. Numerous studies on the impact of ZnO NPs on various biological systems have revealed that ZnO NPs improve seed germination and seedling growth and alter various chemicals in medicinal plants [6,7].

However, little is known about the impacts of the application of ZnO NPs on the bioactive compound contents of *C. longa*. Therefore, the present research aimed to determine the antibacterial properties of a turmeric rhizome extract after it was treated with different doses of ZnO NPs during culture. The research also analyzed the influence of ZnO NPs on the biochemical composition of *C. longa* extract and its relevance to antibiosis.

## 2. Materials and Methods

### 2.1. Plant Materials and Extract Preparation

Turmeric rhizomes were sown into sandy soil according to Khattab, S et al., 2023, and foliar spraying with various concentrations of ZnO NPs (0.0, 5, 10, 20, and 40 mgL^−1^) (catalogue no. 677450, with a particle size of 50 nm, Sigma-Aldrich, St. Louis, MO, USA) was conducted three times during vegetative development at King Faisal University in Saudi Arabia according to [21]. After eight months of culture, the rhizomes generated by various ZnO NP treatments, as well as the control, were collected [21].

Then, 1 g of homogenized air-dried rhizome powder was added to a 28 mL stoppered culture tube and defatted with 30 mL of ethanol for one day while being shaken at 100 rpm on a rotary shaker. The extracts were filtered using a 0.2-micrometer syringe filter.

### 2.2. Determination of Curcumin, Bisdemethoxycurcumin, and Demethoxycurcumin Contents via High-Performance Liquid Chromatography (HPLC)

The curcumin, bisdemethoxycurcumin, and demethoxycurcumin contents of the air-dried *C. longa* rhizome powder, drawn from three plants randomly chosen from each treatment (control and 5, 10, 20, and 40 mgL-ZnO NPs), were assessed according to the methods described by [21]. The presence of curcumin, bisdemethoxycurcumin, and demethoxycurcumin compounds increased in the plants treated with different concentrations of ZnO NP, with enhanced antibacterial results.

### 2.3. Determination of Nanozinc Curcumin Ethanolic Extract Antimicrobial Activity via Disc Diffusion Assay

#### 2.3.1. Disc Diffusion Assay

Bacterial strains of *S. aureus*, *E. coli* ATCC 8739, *Pseudomonas aeruginosa, Actinobacteria baumannii*, and *Bacillus* sp. were obtained from the College of Medicine, King Faisal University. The cultivation of all the bacterial strains was performed in nutrient broth (Sigma Aldrich, St. Louis, MO, USA, Cat. No. 7014), and the disc diffusion assay was performed according to Rad et al., 2021. A total of 100 μL of overnight bacterial culture was inoculated into 10 mL of nutrient broth, and the cultures were grown at 37 °C at a speed of 200 rpm until the turbidity of the culture reached 0.3 at 600 nm [22]. Subsequently, 5 mL of the homogenous bacterial culture of each of the bacterial strains was poured onto individual nutrient agar plates, and each of the plates was gently swirled to ensure that the culture was spread evenly on the agar. In total, eighteen agar plates were inoculated with each bacterial strain. All inoculated plates were left unsealed in the biosafety cabinet to allow excess liquid to absorb into the agar before placing the test solution discs onto the agar. Four discs of 6 mm diameter containing various test solutions were placed onto the agar surface of each inoculated plate. The negative control disc contained a DMSO:water (*v*:*v* 1:1) solution. A positive control disc containing 10 µg of imipenem (cat. no. 7052) was purchased from Condalab, Madrid, Spain. All plates were then incubated at 37 °C overnight before the diameters of the zones of inhibition were measured.

#### 2.3.2. Minimum Inhibitory Concentration Test (MIC)

For the duration of the night, strains were grown in nutrient broth. Curcumin (5 mg/mL) stocks were generated in water after the strains were treated with ZnO NP stocks [23,24].

#### 2.3.3. Time–Kill Test of ZnO NPs

After mixing *C. longa* ZnO NPs with nutrient broth medium containing 1.5 × 10^8^ CFU/mL of bacterial inoculum, the mixture was kept at 37 °C at doses of 0, 0.5, 1, 2, and 4 MIC for *S. aureus* and *P. aeruginosa*. A 0.1 mL medium was grown on Mueller–Hinton agar and incubated at 37 °C for 24 h under different conditions.

### 2.4. Statistical Analyses

The experiment was designed as a complete randomized trial. Two-way analysis of variance (ANOVA) in Statistica 6 (StatSoft Inc., Tulsa, OK, USA) was used to analyze the results of all of the measures. A probability threshold of *p* = 0.05 was used to determine whether or not there was a statistically significant difference in the mean scores between the treatment groups [25]. 

## 3. Results

### 3.1. HPLC Results

High-performance liquid chromatography (HPLC) was used to find out how different doses of ZnO NPs affected the accumulation of bisdemethoxycurcumin, demethoxycurcumin, and curcumin in ethanolic extracts of *C. longa* rhizomes (Figure 1).

When compared to the control treatment, all concentrations of ZnO NPs increased bisdemethoxycurcumin, demethoxycurcumin, and curcumin levels approximately (2.79 and 2.85)-, (2.65 and 2.94)-, and (2.78 and 3.17)-fold, respectively (Figure 2). As the concentrations of ZnO NPs increased, so did the levels of bisdemethoxycurcumin, demethoxycurcumin, and curcumin.

High-performance liquid chromatography (HPLC) was used to find out how different doses of ZnO NPs affected the buildup of bisdemethoxycurcumin, demethoxycurcumin, and curcumin in ethanolic extracts of *C. longa* rhizomes. When compared to the control treatment, all concentrations of ZnO NPs significantly increased bisdemethoxycurcumin, demethoxycurcumin, and curcumin levels (Figure 2).

Compared to the control treatment, spraying the leaves with 5 and 40 mgL^−1^ of ZnO NPs increased the levels of bisdemethoxycurcumin, demethoxycurcumin, and curcumin by about 2.69 and 2.84 times, 2.61 and 3.22 times, and 2.90 and 3.45 times, respectively.

### 3.2. Antibacterial Susceptibility

Figure 3A,B show the outcomes of the agar-disc-diffusion-method-based antibacterial test of the *C. long*-ZnO NPs against *E. coli*, *S. aureus*, and *P. aeruginosa* with MICs of 100 µg/mL. The data showed that, with the exception of *P. aeruginosa* for which the MIC was 100 g/mL, *C. long*-ZnO NPs demonstrated antibacterial efficacy against the examined *S. aureus* and *P. aeruginosa* bacterium strains, but that there was no attempt in *E. coli.* Additionally, the outcomes of the time–kill experiments demonstrated that *P. aeruginosa*, *Actinobacteria baumannii,* and *Bacillus* sp. were killed by ZnO NPs at 4 MIC after 2 h (Figure 3A,C,D), whereas *S. aureus* showed no growth when treated with 4 × MIC of the extract after 6 h (Figure 3B).

### 3.3. The Effect of Nanoparticles on the Antimicrobial Activity of Curcumin Ethanolic Extract

Disc diffusion experiments (Table 1) were used to measure the antibacterial activity against *S. aureus*, *E. coli*, *Pseudomonas aeruginosa*, *Actinobacteria baumannii,* and *Bacillus* sp. The experiments showed that ethanolic fruit extracts of curcumin plants treated with zinc nanoparticles had different levels of antibacterial activity. Plant extracts that had not been treated with zinc nanoparticles (control) were more effective against all pathogenic bacteria tested (*S. aureus*, *E. coli*, *Pseudomonas aeruginosa, Actinobacteria baumannii,* and *Bacillus* sp.). Plant extracts treated with zinc nanoparticles were more effective against *S. aureus*. Ethanolic extracts of plants treated with zinc nanoparticles showed a rise in the curcumin content (Figure 3). This led to improved antibacterial activity against *S. aureus* (nearly on par with that exhibited by 10 g of imipenem), but not against *E. coli* or *Pseudomonas aeruginosa*. Anti-*E. coli* activity was shown to be diminished in plant extracts treated with 5, 10, 20, or 40 ug/L compared to control plant extracts, as shown in Figure 4.

## 4. Discussion

Recently, the different chemical and physical characteristics of metal and diamond nanoparticles were confirmed when they came into contact with biological properties. They have a variety of outcomes on fungus and bacteria. With the potential to modify traditional agricultural practices, conserve fertilizer, and lessen environmental pollution, nanotechnology has a clear place in the future of agriculture and food production. Because of the behavior of microbes, treating a number of bacterial diseases throughout the world may be difficult due to multidrug resistance (MDR). Because of this issue, the scientific community has become more focused on the development of other antibacterial drugs that can function in lieu of *MRSA*. There is a wide selection of well-known antibacterial medications available on the market. However, a significant portion of the world’s populace is unable to benefit from the use of these antibacterial drugs because of their toxicity, high cost, and unfriendly nature towards the environment. In addition, another significant challenge is the development of antibiotic resistance in bacterial populations. Microorganisms that are resistant to many drugs, sometimes known as superbugs, acquire resistance to conventional medicines that are typically used. *Staphylococci* are thought to be the most common organisms responsible for infections associated with biofilms. As a result, the development of novel antibacterial materials that are trustworthy in terms of cost, friendly to the environment, and technologically sophisticated is a potential solution for the treatment of bacterial diseases. One promising approach to combating superbugs is to modulate the metabolic pathways of medicinal plants by applying non-metals in order to boost the production of bioactive compounds. In the current study, the effects of the foliar application of zinc oxide nanoparticles on the concentration of bioactive components, as well as their antibacterial activity against bacterial pathogens, were investigated.

Foliar application of ZnO NPs increased the amount of bioactive compound in *C. longa* plants as compared to the control treatment. This may be caused by the enhanced nutrient uptake efficiency associated with zinc oxide fertilizers, a product containing nanostructures that encourage nutrient uptake by plants while conserving nutrient resources [26,27,28]. As a member of the lactam family of antibiotics, imipenem is effective against both Gram-positive and Gram-negative bacteria, including those that are resistant to MRSA and other antibiotics [29]. Our disc diffusion assay results were in accordance with this, as the imipenem discs exhibited 18,10, and 15 mm inhibition zones against *S. aureus*, *E. coli,* and *Pseudomonas aeruginosa*, respectively. The antimicrobial activity of ethanolic extracts of curcumin treated with zinc nanoparticles against various bacterial species, including *S. aureus, E. coli*, and *Pseudomonas* aeruginosa, has been previously reported [21,30]. In a study of ethanolic extracts by [31], the authors observed larger inhibition zones in plates inoculated with *Pseudomonas aeruginosa* compared to those inoculated with *E. coli.* In contrast, in another study on an ethanolic extract by Rad, Z.M. et al. (2021), the authors observed inhibition zones of similar sizes between plates inoculated with *S. aureus* and *Pseudomonas aeruginosa*, which is consistent with the observation in the current study [22]. The increase in the active component content in the ethanolic extracts of curcumin after its treatment with zinc nanoparticles resulted in an increase in antimicrobial activity against *S. aureus* as well as *Pseudomonas aeruginosa*, but not *E. coli.* Mirzahosseinipour et al. (2021) also observed similar results, whereby increases in the active content in ethanolic extracts treated with zinc nanoparticles had a lower enhancing effect on antimicrobial activity towards *E. coli* compared to that of *S. aureus* and *Pseudomonas aeruginosa*. *C. longa* extracts have been shown to have antibacterial activity against Gram-positive and Gram-negative bacteria, including against those responsible for human illnesses and antibiotic resistance [31,32]. Various antimicrobial mechanisms are suggested for the antimicrobial activities of *C. longa* extracts, including inhibition of bacterial biofilms and quorum sensing, damage to the cell wall and/or cell membrane, interference with cellular processes via DNA and protein targeting, and lipid peroxidation [33,34].

Bacterial biofilms are made up of cell aggregates that connect to an interface or a surface and become enmeshed in a self-produced matrix of extracellular components, such as polysaccharides and proteins [35]. In challenging environments, such as those that include antimicrobial compounds, it is vital for bacteria to develop a biofilm in order to continue their existence [35]. Biofilms may protect bacteria from being killed by antimicrobials. According to the findings of our investigation, the antibiofilm-forming activity demonstrated by *C. longa* extracts was comparable to that of its antibacterial activity. The capacity of bacteria to form biofilms, rather than the bacteria themselves existing as free organisms, has been related to the development of antibiotic resistance in bacteria. Because biofilms have the ability to render antimicrobials ineffective [35], antimicrobial agents cannot reach bacterial cells when biofilms are present. This prevents antimicrobial agents from killing bacteria. The creation of a biofilm is a multistage process that includes adhesion, maturation/proliferation, and separation, as has been shown by a significant body of research. Infections caused by *S. aureus* that are linked with biofilms often entail the establishment of nonspecific antibiotic resistance via biofilms [35]. As *C. longa* extracts are used to inhibit the production of biofilms, their use therefore restricts the development of antibiotic resistance in bacteria.

The antimicrobial activity of as-formed zinc oxide nanoparticles (ZnO NPs) against Gram-positive *Staphylococcus aureus* and Gram-negative *Acinetobacter baumannii* has been investigated. The developing aerobic pathogen *Acinetobacter baumannii* is capable of causing serious infectious diseases. This organism is a member of the Neisseriaceae genus. Due to its pathogenic potential, including its propensity to cling to surfaces, develop biofilms, demonstrate antimicrobial resistance, and collect genetic material from other genera, it is a complex and tough pathogen to manage and remove. *Staphylococcus aureus* (family: *Staphylococcaceae*) is another bacterium that may cause significant illness [36,37,38]. It is responsible for a wide variety of infections, including those of the blood, bones, and joints, as well as pneumonia.

## 5. Conclusions

The potential of ZnO NPs as growth and multiplication promoters for *C. longa* was investigated. These NPs affected the phytochemical profiles of ethanolic extracts produced from rhizomes generated from different ZnO NP treatments. The outcomes showed that ZnO NPs were the best enhancers for *C. longa*’s active compounds. Our HPLC investigations revealed that the phytochemical profiles of ethanolic extracts of *C. longa* plantlets treated with NPs were altered, which was consistent with observations from several plant species. The biologically distinct antibacterial activity of these ethanolic extracts against *Pseudomonas aeruginosa*, *S. aureus*, *E. coli*, *Actinobacteria baumannii,* and *Bacillus* sp. was also a biological reflection of the differences in the phytochemical contents of these extracts. The phytochemical compositions of the ethanolic extracts produced from *C. longa* plantlets changed due to the application of ZnO NPs, and these compositional differences led to the different antimicrobial activities of these extracts. These results demonstrate that ZnO NPs significantly increase bioactive compound production in *C. longa*, a quality that can be exploited in the fight against antibiotic resistance in pathogenic bacterial species.

## Figures and Tables

**Figure 1 molecules-28-04935-f001:**
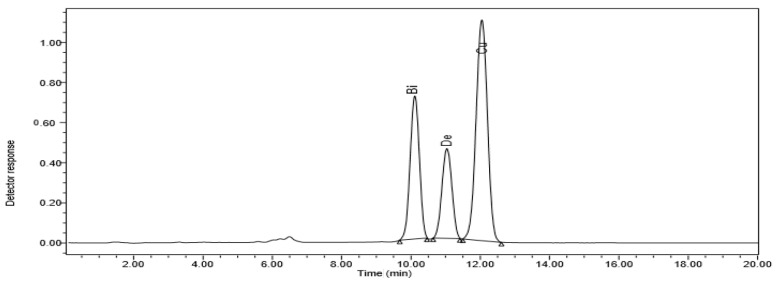
HPLC chromatogram of *C. longa* organic extract subjected to 20 ppm ZnO NPs. The curcuminoids bisdemethoxycurcumin (Bi), dimethoxycurcumin (De), and curcumin (Cu) are displayed.

**Figure 2 molecules-28-04935-f002:**
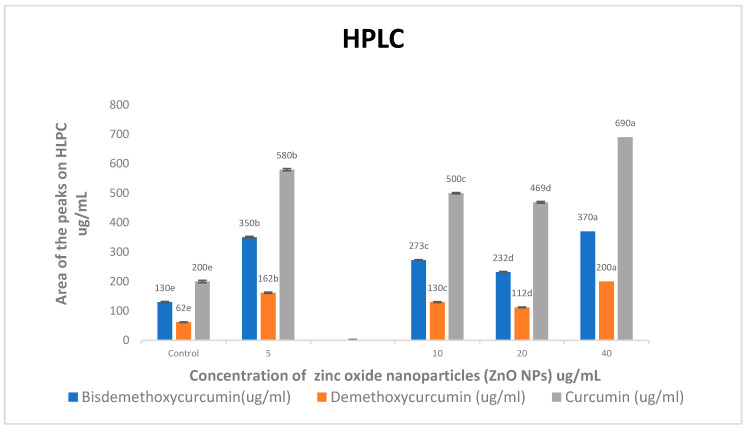
Effects of zinc oxide nanoparticle (ZnO NP) treatments on bisdemethoxycurcumin, demethoxycurcumin, and curcumin (ug/mL) accumulation in *C. longa.* Means followed by the same letter within a column are not significantly different at the 0.05 level of probability according to Duncan’s test.

**Figure 3 molecules-28-04935-f003:**
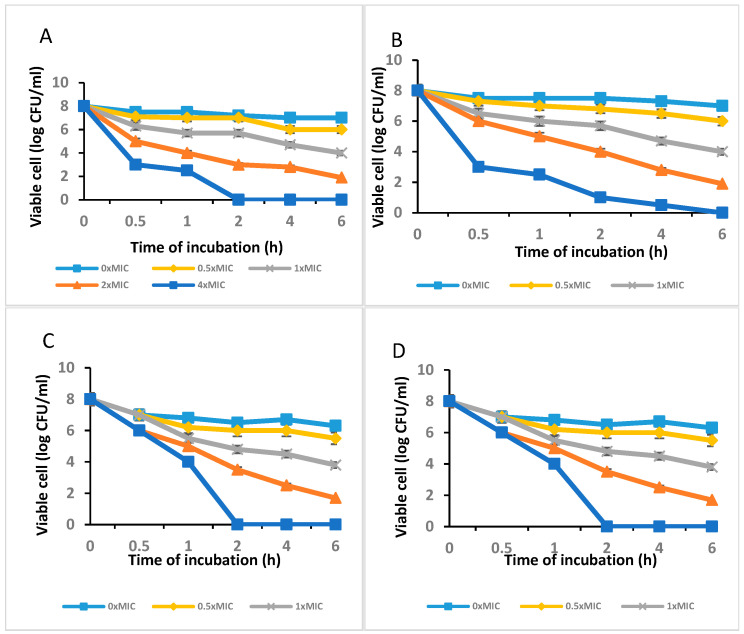
The time–kill curve plots of (**A**) *P. aeruginosa*, (**B**) *S. aureus*, (**C**) *Actinobacteria baumannii*, and (**D**) *Bacillus* sp. after exposure to *C. longa*-ZnO NPs.

**Figure 4 molecules-28-04935-f004:**
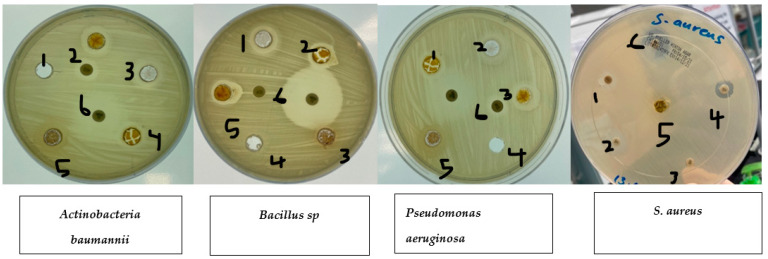
Effects of zinc oxide nanoparticle (ZnO NP) treatments on bacteria. 1. Control. 2. *Curcumin* extract 5 mg/L. 3. *Curcumin* extract 10 mg/L. 4. *Curcumin* extract 20 mg/L. 5. *Curcumin* extract 40 mg/L. 6. Imipenem 10 µg.

**Table 1 molecules-28-04935-t001:** Zone of inhibition (mm) exhibited by various test solutions on plates inoculated separately with *Staphylococcus aureus* (*S. aureus*), *Escherichia coli* ATCC8739 (*E. coli*), *Pseudomonas aeruginosa*, *Actinobacteria baumannii,* and *Bacillus* sp. with ethanolic fruit extracts of curcumin plants treated with zinc nanoparticles. Each of the mean measurements was obtained from the average of 18 replicate plates. The test solution positive control was 10 µg of imipenem. Five curcumin extract test solutions (5 mg/L) were prepared from ethanolic fruit extracts from three individual plants from each treatment and the control. No discs containing the test solution negative control (DMSO: water, *v*:*v*: 1:1) produced an inhibition zone in any of the 18 replicate plates.

Test Solution				Inhibition Zone (mm)	
	*S. aureus*	*E. coli*	*Actinobacteria baumannii*	*Bacillus* sp.	*Pseudomonas aeruginosa*
*Curcumin* extract (control)	3 ± 1 ^f^ *	6 ± 2 ^b^	4 ± 2	5 ± 2	7 ± 2 ^f^
*Curcumin* extract 5 mg/L	12 ± 1 ^b^	5 ± 2 ^c^	8 ± 1	10 ± 3	13 ± 1 ^b^
*Curcumin* extract 10 mg/L	11 ± 1 ^c^	0 ± 0	13 ± 2	14 ± 1	9 ± 1 ^d^
*Curcumin* extract 20 mg/L	7 ± 2 ^e^	0 ± 0	10 ± 1	9 ± 2	8 ± 1 ^e^
*Curcumin* extract 40 mg/L	9 ± 1 ^d^	0 ± 0	8 ± 3	7 ± 2	10 ± 1 ^c^
Imipenem 10 µg	18 ± 3 ^a^	10 ± 2 ^a^	18 ± 2	17 ± 2	15 ± 2 ^a^

* Means followed by the same letter within a column are not significantly different at the 0.05 level of probability according to the L.S.D. test.

## Data Availability

Not applicable.

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
