# Peer review of "Enhancement of the Bioactive Compound Content and Antibacterial Activities in Curcuma Longa Using Zinc Oxide Nanoparticles"

_molecules, 2023, doi:10.3390/molecules28134935_

Round 1

Reviewer 1 Report

The author studied the impact of the ZnO nanoparticles on the improvement of the bioactive compound production in Curcumin longa. He also tested their impact on anti-bacterial activities in Curcuma longa. This work has some merit. However, it should be improved following the reviewer comments:

1/ It is suggested to avoid verbs in the title; it can be “Enhancement of the bioactive compounds content and the anti-bacterial activities in Curcuma longa using Zinc oxide nanoparticles”

2/ In this paper the author used ZnO nanoparticles as antibacterial agent. He should add how to synthesize ZnO nanoparticles and what kind of used process to obtain them, especially ZnO features depend on the growth method.

3/ The author should add the crystallite size of ZnO material.

4/ Is the ZnO nanoparticle size and its morphology affected the antibacterial results?

5/ It is suggested to add some photos to show the zone of inhibition exhibited by various test solutions mentioned in table 1.

Minor editing of English language are required.

Author Response

I did all the requirements according to the comments

Reviewer 2 Report

The paper entitled, “Zinc oxide nanoparticle enhances the bioactive compounds content and the anti-bacterial activities in Curcuma longa”.  Comprises of two aspects, firstly the application of ZnO nano-particles  and  their impact of secondary metabolite concentration and secondly the synthesis of Zn nanoparticles with extracts and their antibacterial role.

I have some observation and suggest following changes/ answers

1.       How were the different concentrations of treatments selected for spray and how these concentrations were maintained while applying treatments?  the complete methodology of plant growth and treatments should be provided. Was any other fertilizer or nutrient or water applied.

2.       Which type of soil was used for the cultivations of rhizome and we cannot  ignore the effect of soil parameters on the growth and secondary metabolites concentrations as no soil analysis was carried out. As suggested above the field experimental design is required.

3.       The methods should be incorporated with Nanozinc Curcumin Ethanolic Extract preparation.

4.       There are many spelling mistakes throughout the manuscript like line 96, 133 and 85, besides follow the uniform pattern while mentioning the scientific names and  italicize.

5.       Some clarity is needed  in table 1 and figures as to whether only the extracts were used on plates  or the  Nanozinc Curcumin Ethanolic Extracts were used.

6.       Discussion part needs improvement.

7.       Lot of scientific names in references are not in italic form.

8.       There are some grammatical mistakes which need to be addressed.

The role of Zn in plants at the lower concentrations is more as a nutrient and stress alleviator; however the higher concentrations are toxic to plants affecting their growth and development. Usually, the production of secondary metabolites is enhanced under stress conditions. This would  mean that zinc concentrations applied to  plants were inducing stress in plants.

grammatical mistakes should be corrected

Author Response

Dear Reviewer, 

I did respond to all the comments 

Regards,

Munirah

Round 2

Reviewer 1 Report

The authors corrected the manuscript and improved it. This work can be considered for publication in Molecules.

Minor editing of English language are required.